# The lncRNA41584-miR3047-z-*BmCDK20* ceRNA Regulatory Network Influences Reproductive Development in Male Silkworms (*Bombyx mori*)

**DOI:** 10.3390/insects16111120

**Published:** 2025-11-01

**Authors:** Tianchen Huang, Juan Sun, Shanshan Zhong, Dongxu Shen, Heying Qian, Qiaoling Zhao

**Affiliations:** 1College of Biotechnology, Jiangsu University of Science and Technology, Zhenjiang 212100, China; 210111801106@stu.just.edu.cn (T.H.); sj5062884263@163.com (J.S.); shan853@foxmail.com (S.Z.); shendongxu0311@163.com (D.S.); qhysri@just.edu.cn (H.Q.); 2Key Laboratory of Silkworm and Mulberry Genetic Improvement, Ministry of Agriculture and Rural Affairs, Sericultural Scientific Research Center, Chinese Academy of Agricultural Sciences, Zhenjiang 212100, China

**Keywords:** lepidopteran, male sterility, spermatogenesis, lncRNA, cell proliferation

## Abstract

**Simple Summary:**

Tissue-specific long non-coding RNAs (lncRNAs) are emerging as important biomarkers for biological traits. This study investigated a testis-specific lncRNA, lncRNA41584, which was previously found to be under-expressed in male-sterile silkworm mutants. While its association with sterility was clear, the underlying mechanism remained unknown. We discovered that lncRNA41584, primarily located in the cytoplasm, functions as a molecular sponge for a small RNA called miR-3047-z. This interaction indirectly boosts the levels of a key cell cycle protein, BmCDK20 (a cyclin-dependent kinase). Experiments in silkworm cells showed that disrupting this lncRNA41584/miR-3047-z/*BmCDK20* pathway impaired cell multiplication, causing most cells to halt in the early stage of cell division (G1 phase). Further studies in live silkworms confirmed that blocking lncRNA41584 or increasing miR-3047-z led to abnormally shaped sperm bundles and significantly reduced fertilization success. Our findings reveal that these three molecules form a coordinated regulatory network (a competing endogenous RNA, or ceRNA, axis) essential for proper sperm production. This work provides fundamental insights into the causes of male infertility in insects and identifies potential new targets for developing insect sterility-based pest control strategies, offering a more environmentally friendly approach to pest management.

**Abstract:**

Background: Tissue-specific long non-coding RNAs (lncRNAs) represent potential biomarkers. The testis-enriched lncRNA41584, previously identified as downregulated in male-sterile silkworm mutants (*JMS*, *GMS*), is associated with male sterility, but its functional mechanism remained unknown. Subcellular localization, dual-luciferase reporter assays, MTT, and flow cytometry were employed to examine lncRNA41584–miR-3047-z–*BmCDK20* interactions. In vivo functional validation included lncRNA41584 knockdown and miR-3047-z overexpression in *Bombyx mori*. lncRNA41584 localizes predominantly to the cytoplasm and acts as a competing endogenous RNA (ceRNA) by sponging miR-3047-z, thereby upregulating the cyclin-dependent kinase *BmCDK20*. Perturbation of this axis impaired BmN cell proliferation, causing G1 phase arrest, and led to spermatocyst malformation, reduced fertilization rates, and increased unfertilized eggs. The lncRNA41584–miR-3047-z–*BmCDK20* ceRNA network is essential for testicular cell cycle progression and spermatogenesis in silkworms, offering mechanistic insights into lepidopteran male sterility and potential targets for pest fertility regulation.

## 1. Introduction

During reproductive development, the *Bombyx mori* Linnaeus, 1758 generates two types of sperm: nucleated eupyrene sperm and anucleated apyrene sperm [1]. These two sperm morphologies play distinct critical roles in fertilization. Specifically, nucleated eupyrene sperm facilitate oocyte fertilization, while spermatogenesis is essential for the formation of nucleated eupyrene sperm. During spermatogenesis, germ cells in the silkworm testis undergo rapid proliferation and become encapsulated by cytoplasmic filaments to form spermatocytes. Recent studies on silkworm male sterility have revealed that failures in specific spermatogenesis processes can lead to male infertility. Chen demonstrated that *Bmmael* deficiency in silkworms causes spermatogenic defects during the elongation phase, resulting in nuclear condensation arrest and subsequent male sterility [2]. Additionally, Yang identified *BmPrmt5* and *BmVasa* as critical factors for silkworm spermatogenesis. These genes regulate meiotic division in testicular cells, ultimately impairing male reproductive capacity in silkworms [3].

In recent years, in-depth studies on lncRNAs have revealed their potential as biomarkers. Compared with mRNAs, lncRNAs exhibit significant tissue and environmental specificity and play important roles in gene expression regulation, development, and immune responses. lncRNAs can regulate a series of key cellular processes by modulating gene expression, and during spermatogenesis, they affect sperm production by influencing cell proliferation, differentiation, or apoptosis, which in turn impacts reproductive development and leads to infertility [4]. Overexpression of lncRNA-Gm2044 leads to abnormalities in acrosome assembly and sperm plasma membrane processes, inhibits Utf1 translation, affects the transformation and meiosis of germ cells, and blocks spermatogenesis [5]; SpgalncRNA1 and Spga-lncRNA2 play important roles in maintaining the stemness of spermatogonia [6]; LncRNA033862 plays a key role in GDNF-mediated spermatogonial stem cell self-renewal [7]; Mrhl interacts with other regulatory factors such as the Myc-Max-Mad, Sox8 promoter corepressor Sin3a, and coactivator Pcaf, and regulates spermatogonial differentiation [8]; In the nucleus, the lncRNA NLC1-C suppresses the transcription of miR-320a and miR-383, and promotes the proliferation of testicular embryonal carcinoma cells by binding with Nucleolin [9]; Testis-enriched lncRNA-AK015322, highly expressed in spermatogonial stem cells, promotes their proliferation [10]; lncRNA H19 regulates the proliferation and apoptosis of male germline stem cells [11]; In addition, several lncRNAs (Drm, HongrES2, Tsx, LncRNA-tcam1, Tug1, Tesra, and AK015322) have been functionally verified for their roles in spermatogenesis [4].

The cytoplasm is the location for the presence and action of many lncRNAs. Cytoplasm-localized lncRNAs play roles in regulating mRNA stability, controlling mRNA translation, acting as competitive endogenous RNAs (ceRNAs), serving as microRNA precursors, and mediating protein modifications [12]. Among these, the concept of “competitive endogenous RNA” (ceRNA) forms regulatory networks in the transcriptome, controlling the cellular processes of key genes and significantly expanding the functional genetic information in the genome [13]. ceRNA theory suggests that mRNA transcription levels are regulated through ceRNAs and that microRNAs are regulatory factors for gene expression that reduce the stability or limit the translation of target RNAs [14], effectively controlling mRNA transcription levels. Both coding and non-coding RNA targets can interfere with each other by competing for microRNA binding. The lncRNA AK01522 inhibits the effect of miR-19b-3p on ETV5 expression, thereby promoting SSC proliferation [10]. CBR3-AS1 regulates hsa-miR-145-5p, which affects the MAP3K5 pathway, closely related to iron metabolism in benign prostatic hyperplasia [15]. WDFY3-AS2 inhibits miR-21-5p/miR-221-3p/miR-222-3p, regulating TIMP3 and affecting the occurrence of renal clear cell carcinoma [16]. LncRNA H19, as a ceRNA of let-7 g, promotes the conversion of endothelial cells to mesenchymal cells in hypoxic pulmonary arterial hypertension by regulating the TGF-β signaling pathway [17]. MEG3-miR-21-SPRY1-NF-kB may participate in the proliferation, autophagy, and apoptosis of spermatogenic cells, leading to azoospermia [18]. As research on lncRNAs continues, the validity of ceRNA regulatory mechanisms is further confirmed.

Cell division and the coordination of cell growth ensure cellular size homeostasis, whereas cyclin-dependent kinases (CDKs), as basic mitotic drivers, ultimately ensure this size homeostasis [19]. Some members of the *CDK* family, such as the *Cdkn2aip* gene, are crucial for spermatogenesis and the development of germ cells. Their deletion leads to approximately 19% synaptic failure in spermatocytes, and their in vitro deletion leads to prolonged S phase, increased DNA damage, and apoptosis [20]. The CDK family member CDK20 binds to the ubiquitin ligase Kelch-like ECH-associated protein 1 (KEAP1), and *CDK20*-deficient cells exhibit impaired cell proliferation and G2/M arrest defects [21]; *AccCDK20* is involved in regulating various stress response and DNA damage repair pathways [22]; and *CCRK/CDK20* interacts with *BROMI/TBC1D32* and regulates the hedgehog signaling pathway in cilia [23]. These findings highlight the importance of the CDK20 protein in cell mitosis, and its dysfunction may lead to abnormal cell proliferation.

As a model insect of the Lepidoptera order, the silkworm (*Bombyx mori*) serves as an ideal biological model, and its special mutants have become excellent materials for modern genetic engineering research. The silkworm genetic male sterile mutant (*JMS*) is a natural mutant characterized by an abnormal cocoon shape and male sterility in homozygous males. In previous studies, we conducted whole-transcriptome sequencing of testes from *JMS* and the wild-type silkworm strain *JN8HUA*, identifying 205 DElncRNAs [24]. In this study, we further explored the significantly downregulated lncRNA41584 in *JMS* and found that it is localized in the cytoplasm, where it regulates miR-3047-z through a ceRNA network and further modulates *BmCDK20* (SilkDB 3.0 ID: BMSK0014768), leading to abnormal testicular cell proliferation and ultimately affecting reproduction. This study reveals the role of lncRNAs in male silkworm reproduction and provides new insights into the potential mechanisms of male sterility. It also offers new targets for insect sterility technologies, opening new avenues for pest control.

## 2. Materials and Methods

### 2.1. Silkworm Breeding and BmN/HEK293T Cell Culture

*Bombyx mori JN8HUA*, *JMS*, and p50 were preserved at Jiangsu University of Science and Technology. *B. mori* were fed at 25 ± 1 °C and 80 ± 5% relative humidity under a 12 h light/dark cycle. Male and female moths were allowed to mate for 4–6 h at 25 °C under light conditions, and the female moths laid eggs in the dark. After egg laying, the eggs were incubated at 25 °C for 8–10 days, and the fertilization rates were determined. BmN cells were cultured at 27 °C in TC-100 medium containing 100 μg/mL penicillin, 100 μg/mL streptomycin, and 10% fetal bovine serum. HEK293T cells were cultured in high-glucose DMEM supplemented with 10% fetal bovine serum (FBS), 100 μg/mL penicillin, and 100 μg/mL streptomycin at 37 °C with 5% CO_2_. All samples were prepared in three independent experiments, with each sample being repeated three times.

### 2.2. Total RNA Extraction and Quantitative Real-Time RT–PCR (RT–qPCR)

Total RNA was extracted via TRIzol^®^ reagent (Invitrogen, Carlsbad, CA, USA). Total RNA was reverse transcribed into cDNA via the Evo M-MLV Reverse Transcription Premix Kit (containing gRNA removal reagent for qPCR) Ver. 2 (Accurate Biotechnology, Changsha, China). The synthesized cDNA was diluted to 200 ng/µL as the template for qPCR. A SYBR^®^ Green Pro TaqHS Premixed qPCR Kit (containing tracking dye) was used to prepare the reaction mixture (5 μL of 2 × SYBR^®^ Green Pro TaqHS Premix, 0.4 μL of forward primer, 0.4 μL of reverse primer, 3.2 μL of ultrapure water, and 1.0 μL of template) for qPCR via the Light Cycler^®^ 96 SW 1.1 system (Roche, Rochester, NY, USA).

MicroRNA Transcription Detection: A miRNA 1st Strand cDNA Synthesis Kit (stem–loop) (Vazyme, Nanjing, China) was used to reverse transcribe total RNA templates into cDNA, and a SYBR^®^ Green Pro TaqHS Premixed qPCR Kit was used to detect miRNA transcription levels.

### 2.3. Full-Length Amplification of lncRNA41584

3′ RACE: Nested PCR was used. Total RNA was extracted from fresh testis tissue of p50 fifth-instar silkworms on day 3 and reverse transcribed into cDNA via an Aicor reverse transcription kit. The 3′ end-specific primer 41584-GSP and long primer were used for amplification, and the PCR product was then used as a template for amplification with the 3′ end-specific primer 41584-GSP-2 and short primer. After gel extraction and purification (CW2302M, CoWin Biosciences, Taizhou, China), the product was ligated to the universal vector pMD-19T via a Takara 19T Kit and sequenced by Sangon Biotech (Shanghai, China).

5′ RACE: The SMARTer^®^ RACE 5′/3′ Kit User Manual was used to reverse transcribe total RNA from fresh testis tissue of p50 fifth-instar silkworms into RACE-ready cDNA. The RACE-ready cDNA was used as a template, with the upstream primer being the lncRNA-specific GSP primer and the downstream primer being the 10 × UPM random primer from the SMARTer^®^ RACE 5′/3′ Kit. Touchdown PCR was used to amplify the 5′ end sequence. The PCR products were gel extracted, purified, ligated to the pMD19-T-Simple vector (Takara, Nanjing, China) and sequenced by Sangon Biotech (Shanghai, China). The touchdown PCR annealing temperature was gradually reduced from 62 °C to 58 °C, with 5 cycles of denaturation, annealing, and extension at 62 °C and 60 °C, followed by 20 cycles at 58 °C.

Full-Length Fragment Acquisition: By analyzing the 3′ and 5′ end sequencing sequences of the RACE products, corresponding primers were synthesized to amplify the full-length cDNA via the use of RACE-ready cDNA as a template.

### 2.4. Subcellular Localization of lncRNA41584 in Testicular Cells

#### 2.4.1. Fluorescence In Situ Hybridization (FISH)

Testis tissue from p50 fifth-instar silkworms on day 3 was prepared into paraffin sections, stained, denatured, and blocked for prehybridization. A Cy3-labeled specific RNA probe, 41584-cy3, was used, with a meaningless sequence used as the negative control (NC) and 18S as the cytoplasmic positive control. After hybridization and washing, the nuclei were counterstained with DAPI. Images were observed and collected via a Nikon fluorescence microscope. The UV excitation wavelength was 330–380 nm, the emission wavelength was 420 nm (blue light), the Cy3 red light excitation wavelength was 510–560 nm, and the emission wavelength was 590 nm (red light).

#### 2.4.2. Nuclear–Cytoplasmic RNA Separation

A Cytoplasmic and Nuclear RNA Purification Kit (Norgen, New Orleans, NY, USA) was used to extract nuclear and cytoplasmic RNA according to the kit instructions. Actin and U6 were used as references to verify the separation efficiency, and qRT–PCR was performed to detect the transcription levels of lncRNA41584.

### 2.5. Tissue and Temporal Expression Profiles of lncRNA41584

To investigate whether lncRNA41584 exhibits tissue-specific expression patterns, we collected 11 tissues from p50 fifth-instar silkworms on day 3, including silk glands, fat bodies, nerve cords, midguts, hemolymph, respiratory clusters, heads, Malpighian tubules, ovaries, testes, and epidermis. Testis tissue from p50 fifth-instar silkworms on days 1, 2, 3, 5, and 2 of the spinning period was also collected. After total RNA was extracted, cDNA was synthesized, and qRT–PCR was performed to analyze the transcript levels of lncRNA41584.

### 2.6. RNAi and Overexpression of lncRNA41584, miR-3047-z, and BmCDK20 in the BmN Cell Line

Small interfering RNAs (siRNAs) for inhibiting lncRNA41584 and *BmCDK20* expression, as well as mimics and inhibitors of miR-3047-z, were synthesized by GenePharma (Suzhou, China).

lncRNA41584 was transfected into BmN cells via the GP-transfect-Mate system (GenePharma, Suzhou, China). BmN cells were transfected with 80 pmol of siRNA via the liposome transfection method in 12-well plates (70% confluence). After 48 h of transfection, RNA was extracted via TRIzol (Takara, Dalian, China). After 48 h of transfection, qRT–PCR was performed to determine the silencing efficiency of lncRNA41584 or *BmCDK20*, with *BmGAPDH* used as the internal control. The changes in the expression of miR-3047-z were normalized to those of U6.

The complete lncRNA41584 sequence was cloned and inserted into the plasmid pIZT-mercherry-v5-his (provided by the Biotechnology Laboratory, Jiangsu University of Science and Technology, Zhenjiang, China) at the restriction sites BamH1 and EcoR1. The plasmid was extracted via an endotoxin-free plasmid extraction kit (Omega, Norcross, GA, USA). Then, 2 μg of the pIZT-mercherry-v5-his-lncRNA41584 plasmid was transfected into BmN cells (70% confluence). After 48 h of transfection, RNA was extracted via TRIzol (Takara, Dalian, China). After 72 h of transfection, RT–qPCR was performed to determine the overexpression efficiency of lncRNA41584, as well as the transcription levels of miR-3047-z, *BmCDK20*, and *BmPCNA*. *BmGAPDH* was used as the internal control, and U6 was used for miR-3047-z.

### 2.7. MTT Assay for Detecting Cell Proliferation

An MTT Cell Proliferation and Cytotoxicity Assay Kit (Beyotime, Shanghai, China) was used to detect cell proliferation in the BmN cell line following overexpression and RNAi treatments. The specific procedure was as follows: 100 µL (approximately 2000 cells) was added to each well of a 96-well plate. After transfection with the overexpression vectors, RNAi, and corresponding controls, 10 µL of MTT solution was added to each well at 24 h, 48 h, and 72 h posttransfection. The cells were incubated for an additional 4 h, followed by the addition of 100 µL of formazan dissolving solution to each well. The plates were gently mixed and incubated until the formazan had completely dissolved. The absorbance was measured at 570 nm.

### 2.8. Flow Cytometry Analysis of the Cell Cycle at 48 h After Knockdown of lncRNA41584 and BmCDK20

A Cell Cycle Assay Kit (red fluorescence) (Elabscience^®^ Biotechnology Co., Ltd., Wuhan, China) was used. After 48 h of lncRNA41584 and *BmCDK20* knockdown, the cells were washed with PBS, treated with anhydrous ethanol, mixed thoroughly, and stored at −20 °C for 1 h or overnight. The cells were centrifuged at 300× *g* for 5 min, the supernatant was discarded, and the cells were resuspended in PBS for 15 min at room temperature. After a second centrifugation at 300× *g* for 5 min, the supernatant was discarded, and 100 µL of RNase A reagent was added to suspend the cells. The cells were incubated in a 37 °C water bath for 30 min. Then, 400 µL of PI reagent (50 µg/mL) was added, mixed thoroughly, and incubated for 30 min at 2–8 °C in the dark. The fluorescence intensity was immediately measured with a flow cytometer at an excitation wavelength of 488 nm for red fluorescence.

### 2.9. Dual-Luciferase Reporter Assay

TargetScan (https://www.targetscan.org/vert_80/, accessed on 10 April 2025) and RNAhybrid (https://bibiserv.cebitec.uni-bielefeld.de/rnahybrid/, accessed on 10 April 2025) were used to predict the targets of miR-3047-z-lncRNA41584 and miR-3047-z-*BmCDK20*. Among the predicted targets, the miR-3047-z sequence was found to match the 3′ untranslated region (UTR) of *BmCDK20-well*. The dual-luciferase reporter system was used to detect the interaction between a miRNA and its target gene. A luciferase reporter vector for the target gene was constructed. Total RNA was extracted from 5-day-old silkworm testis tissue, and cDNA was synthesized via the PrimeScript™ II First Strand cDNA Synthesis Kit (TaKaRa, Dalian, China). The Fastpfu high-fidelity enzyme (Fullgold, Beijing, China) was used to amplify the 472 bp 3′ UTR sequence of *BmCDK20* (primers in Table 1). The 627 bp region of lncRNA41584′s third exon containing the predicted binding site of miR-3047-z was also amplified. The *BmCDK20* 3′ UTR sequence and the third exon of the lncRNA41584 PCR products were digested with SacI and XhoI and ligated into the pMD19-T-Simple vector (TaKaRa, Nanjing, China). After SacI and XhoI digestion of the recombinant vector, it was ligated into the pmirGLO dual-luciferase reporter vector (Promega, Madison, WI, USA). The resulting vectors were named the pmirGLO-*BmCDK20*-UTR mutant and pmirGLO-lncRNA41584 mutant. The interaction region of the *BmCDK20* 3′ UTR with miR-3047-z was mutated via the primer sequences in Table 1, and the mutations were screened after transformation into *Escherichia coli* Migula, 1895 DH5α competent cells.

The interaction of miR-3047-z with *BmCDK20* was studied in four groups of HEK293T cells: (1) pmirGLO-*BmCDK20*-UTR, (2) pmirGLO-*BmCDK20*-UTR and NC (negative control miR-3047-z mimics synthesized by GenePharma, Shanghai, China), (3) pmirGLO-*BmCDK20*-UTR and miR-3047-z mimics, and (4) pmirGLO-*BmCDK20*-UTR mutant and miR-3047-z mimics. Additionally, the interaction of miR-3047-z with lncRNA41584 was studied in four groups of HEK293T cells: (1) pmirGLO-lncRNA41584, (2) NC and pmirGLO-lncRNA41584, (3) miR-3047-z mimics and pmirGLO-lncRNA41584, and (4) pmirGLO-lncRNA41584 mutant and miR-3047-z mimics. The transfection amount for each miRNA or NC mimic was 100 nmol/L, and the vector concentration was 100 ng per well. After 48 h of transfection, cell lysates were prepared, and luciferase activity was measured via a dual-luciferase assay kit (Yisheng, Shanghai, China). The relative firefly luciferase activity was normalized to Renilla luciferase activity. Three independent experiments were performed, with each sample repeated three times.

### 2.10. Knockdown of lncRNA41584 and Overexpression of miR-3047-z in B. mori

Two micrograms of lncRNA41584 siRNA or miR-3047-z mimics (each) were injected into the gonad region of p50 fifth-instar silkworm larvae on days 3 and 7, as well as on the 8th day of the pupal stage, to induce expression knockdown. Forty-eight hours after injection, testis tissue was collected. Tissue sections of the testes were fixed in 4% paraformaldehyde at room temperature and sent to Jiangsu Aidisheng Biological Technology Co., Ltd. (Yancheng, China) for section scanning. RNA was extracted via TRIzol, and expression levels were detected via RT–qPCR. Changes in the expression of lncRNA41584, *BmCDK20*, and the cell cycle-related gene *BmPCNA* were measured using BmGAPDH as the internal reference gene (Table 1). A miRNA 1st Strand cDNA Synthesis Kit (stem–loop) (Vazyme, Nanjing, China) was used to reverse transcribe total RNA templates into cDNA, and changes in miR-3047-z expression were detected. Three independent experiments were performed, with each sample containing 10 individuals.

### 2.11. Tissue Paraffin Sectioning

After collection, the tissues were dehydrated, embedded, and sectioned into 4 μm thick slices. The paraffin sections were deparaffinized, rehydrated, and stained with hematoxylin for nuclei and eosin for the cytoplasm. Finally, the sections were dehydrated and mounted with neutral balsam for microscopic examination. Nuclei were stained blue, and the cytoplasm was stained red.

## 3. Results

### 3.1. Full-Length Amplification of lncRNA41584

LncRNA41584 is located in the silkworm genome on chromosome 7 and is an antisense non-coding RNA (Figure 1a). After the 3′ and 5′ end sequences of the RACE products were analyzed, corresponding primers were synthesized to amplify the full-length cDNA via the use of RACE-ready cDNA as the template (Figure 1b). The full-length sequence was then sequenced, and gel electrophoresis revealed two distinct bands. The sequencing results revealed various forms of deletions and insertions between the second and third exons of lncRNA41584, and four splice variants were identified, uploaded to NCBI, GENE ID: BankIt2930851. Subsequent experiments used a sequence consistent with GENG ID LOC105841584.

### 3.2. Temporal and Spatial Expression Profiles of lncRNA41584

qPCR of 11 tissues from p50 fifth-instar silkworms on day 3 revealed that lncRNA41584 was specifically expressed in the testis, with lower expression levels detected in the ovaries and trachea (Figure 2a). The tissue specificity of lncRNA41584 suggests that it may play a role in testis development or reproduction in silkworms. LncRNA41584 exhibited stable transcription from the fifth-instar larvae to the second day of spinning, with consistent transcription levels observed (Figure 2b).

### 3.3. Subcellular Localization of lncRNA41584

FISH analysis of the subcellular localization of lncRNA41584 revealed that it is primarily localized in the cytoplasm, with no significant overlap with the nucleus, confirming that lncRNA41584 is present mainly in the cytoplasm (Figure 3a). Nuclear–cytoplasmic RNA separation from silkworm testis tissue revealed that U6 was present mainly in the nucleus, whereas Actin was present primarily in the cytoplasm. The RT–qPCR results revealed significant separation between the nuclear and cytoplasmic fractions of p50 silkworm testis tissue (Figure 3b). Further analysis of the nuclear and cytoplasmic RNA after reverse transcription confirmed that lncRNA41584 is predominantly located in the cytoplasm (Figure 3c).

### 3.4. Prediction of the ceRNA Regulatory Network of lncRNA41584 and Its Transcriptional Level in JMS

On the basis of whole-transcriptome data, the ceRNA regulatory network of lncRNA41584 was predicted to be lncRNA41584-miR-3047-z-BmCDK20. RNAhybrid and TargetScan prediction revealed that lncRNA41584 binds with miR-3047-z (Figure 4a), and the miR-3047-z sequence closely matches the 3′ UTR of the cyclin-dependent kinase BmCDK20 (Figure 4b), suggesting that BmCDK20 might be a potential target of miR-3047-z. This interaction was further validated via a dual-luciferase assay. Compared with the negative control, the luciferase activity of HEK293T cells cotransfected with pmirGLO-lncRNA41584 and miR-3047-z mimics (Figure 4c) and pmirGLO-BmCDK20-3′UTR and miR-3047-z mimics was decreased (Figure 4d), and the luciferase activity was restored after point mutation of the binding site, confirming that BmCDK20 is the actual target of miR-3047-z.

The quantitative results from silkworm mutants (*JMS*) and wild-type JN8HUA at 48 h postpupal stage revealed that, in JMS, lncRNA41584 and *BmCDK20* were significantly downregulated, whereas miR-3047-z was significantly upregulated (Figure 5). These findings suggest that lncRNA41584 regulates *BmCDK20* through miR-3047-z. *BmCDK20* is associated primarily with cell proliferation. Additionally, the proliferation cell nuclear antigen (*PCNA*) was also significantly downregulated, indicating its involvement in DNA replication and processing. *PCNA* serves as a platform mediating protein–DNA interactions, with many proteins, including those involved in cell cycle regulation and DNA processing, binding to *PCNA*.

### 3.5. Impact of lncRNA41584/miR-3047-z/BmCDK20 on BmN Cell Proliferation

To investigate how lncRNA41584 interacts with miR-3047-z and *BmCDK20* in silkworms, siRNA targeting lncRNA41584 was used to knock down lncRNA41584 expression, and the pIZT vector was used to overexpress lncRNA41584 in BmN cells. After 48 h of siRNA transfection and 72 h of overexpression, RT–qPCR was used to analyze the expression of miR-3047-z, *BmCDK20*, *BmCDK2* and *BmPCNA*. In lncRNA41584-knockdown BmN4 cells, miR-3047-z transcript levels were significantly increased, whereas *BmCDK20* mRNA levels were significantly decreased (Figure 6a,b). Moreover, *BmCDK2* and *BmPCNA* gene expression was downregulated, indicating that lncRNA41584 inhibition promoted miR-3047-z expression, suppressed the expression of *BmCDK20* and *BmCDK2*, and affected cell proliferation. In contrast, the overexpression of lncRNA41584 in BmN cells led to the downregulation of miR-3047-z and the upregulation of *BmCDK20*, *BmCDK2* and *BmPCNA* (Figure 6c,d). To further study whether lncRNA41584 acts as a “sponge” or decoy ceRNA for miR-3047-z, miR-3047-z mimics and inhibitors were transfected into lncRNA41584-overexpressing BmN4 cells. A significant increase in miR-3047-z levels decreased *BmCDK20* mRNA transcription levels, and *BmCDK2* and *BmPCNA* gene expression was downregulated (Figure 6e,f). Increased miR-3047-z levels also reversed the increase in the transcript level of lncRNA41584. Conversely, after the significant reduction in the miR-3047-z level, both the *BmCDK20* and lncRNA41584 levels were significantly increased. These results demonstrate that lncRNA41584 regulates *BmCDK20* through miR-3047-z, affecting the silkworm cell cycle and proliferation.

MTT assays revealed that the overexpression of lncRNA41584 promoted cell proliferation, whereas its knockdown significantly inhibited it. Conversely, the knockdown of miR-3047-z promoted cell proliferation, whereas its overexpression inhibited it. *BmCDK20* knockdown also significantly inhibited cell proliferation (Figure 7).

Using the ability of propidium iodide (PI) to bind to intracellular DNA, which varies across different phases of the cell cycle, flow cytometry was used to detect fluorescence intensity to assess changes in the cell cycle. Flow cytometry analysis of lncRNA41584 and *BmCDK20*-knockdown cells revealed that most cells were arrested in the G1 phase (Figure 8).

### 3.6. Knockdown of lncRNA41584 or Overexpression of miR-3047-z Leads to Abnormal Spermatogenesis and Reduced Male Fertility in B. mori

Cell division and growth coordination ensure cellular size homeostasis, with cyclin-dependent kinases (CDKs) serving as key drivers of mitosis to maintain homeostasis. To investigate how lncRNA41584 interacts with miR-3047-z and *BmCDK20* in *B. mori*, lncRNA41584 siRNA was injected into the gonadal region of p50 silkworm larvae on days 3 and 7, and miR-3047-z mimics were injected into the gonads on day 7. Since the fifth instar is a crucial period for spermatogenesis, both nucleated and enucleated sperm undergo morphological changes during this time. The testis tissue was dissected and examined for developmental status. At 48 h after the injection of lncRNA41584 siRNA on the p50 third day in fifth-instar larvae, the expression levels of lncRNA41584 and *BmCDK20* were significantly decreased, whereas the miR-3047-z expression levels were significantly increased (Figure 9a). In the NC group, the testes were filled with mature sperm released from spermatocysts, with only a few spermatocysts (Figure 10a). In the lncRNA41584 siRNA group, testes contained mainly spermatocysts (Figure 10b). After 48 h of injecting lncRNA41584 siRNA and miR-3047-z mimics into p50 silkworms on day 7, the expression levels of lncRNA41584 and *BmCDK20* were significantly decreased, whereas the miR-3047-z transcription levels were significantly increased (Figure 9b). RT–qPCR analysis of *BmPCNA* and *BmCDK2* gene expression in the aforementioned experiments revealed that after lncRNA41584 siRNA injection into p50 silkworms on days 3 and 7, miR-3047-z was upregulated, and *BmCDK20* expression levels decreased (Figure 9a,b). The overexpression of miR-3047-z led to the downregulation of both lncRNA41584 and *BmCDK20* expression.

To investigate the roles of lncRNA41584 and miR-3047-z during late-stage spermatogenesis, a subset of fifth-instar day 7 larvae were subjected to lncRNA41584 knockdown for 48 h, while another group received miR-3047-z overexpression treatment for the same duration. Testicular tissue sections were then collected from these specimens. In the Figure, 1 represents primary spermatocytes, 2 represents spermatocysts during spermatogenesis, 3 represents spermatogonia, and 2′ represents abnormal spermatocysts during spermatogenesis. Compared with the NC-injected control group, the experimental group presented many malformed spermatocysts in the testis tissue (Figure 11a–c).

After lncRNA41584 was knocked down via siRNA-mediated knockdown of lncRNA41584 on the seventh day of the fifth-instar larval stage of p50, the mating and egg laying behaviors of lncRNA41584-knockdown p50 male moths and normal female moths, as well as NC-injected p50 male moths and normal female moths, were observed. Compared with those in the control group, the fertilization rates in the experimental groups were significantly lower, and the number of unfertilized eggs was greater. Inhibition of lncRNA41584 did not cause malformed pupa phenotypes but led to reduced fertility (Figure 12).

## 4. Discussion

Through transcriptome sequencing of the genetic male sterile mutant silkworm GMS and the JMS transcriptome, we identified the common differentially expressed lncRNA41584, which exhibits testis tissue-specific expression. Using RACE technology, we amplified the full-length lncRNA41584, confirming that its full transcript does not encode a protein and predicting its possible open reading frame, excluding the possibility that it expresses small peptides. FISH and nuclear–cytoplasmic RNA separation revealed that lncRNA41584 is predominantly localized in the cytoplasm, where it likely plays roles in regulating mRNA stability and translation, acts as a competitive endogenous RNA (ceRNA), and serves as a precursor to microRNAs.

lncRNAs, as ceRNAs, regulate miRNAs and control the expression of downstream target genes [13]. Numerous studies have shown that lncRNAs affect reproductive processes such as cell proliferation, spermatogenesis, and meiosis. For example, NLC1-C binds to the RNA-binding domain of nucleolin to inhibit the transcription of miR-320a and miR-383, promoting the proliferation of spermatogonia and spermatocytes [9]. In rat testes, HongrES2 was found to be a precursor of mil-HongrES2, and its low expression promotes the process of epididymal sperm maturation [26]. Tsx is highly expressed in primary spermatocytes and plays a role in regulating meiosis [27]. Drm regulates Dmrt1 and may be involved in the transition between mitosis and meiosis during germ cell development [28]. LncRNA-Tcam1 is believed to play a key role in meiosis and regulate immune-related genes in mouse germ cells [29]. Tesra, as a transcriptional activator of the Prss/Tessp genes, is crucial for meiosis in reproductive cells [30]. Gm2044 affects germ cell conversion and meiosis by inhibiting Utf1 translation [5]. On the basis of the GO analysis of DEmRNAs predicted by the JMS transcriptome ceRNA network, the majority of DEmRNAs were enriched in the biological process pathway related to cellular processes. KEGG analysis revealed that DEmRNAs associated with the cell cycle were also prominently enriched [24]. The dysregulation of cellular processes may be a significant factor contributing to male sterility in JMS, and this dysregulation is likely closely related to ceRNA regulatory mechanisms. Our study also suggested that the lncRNA41584-miR-3047-z-*BmCDK20* ceRNA regulatory network may play a key role in the male sterility traits observed in the GMS and JMS mutants. In the silkworm BmN cell line, we found that lncRNA41584 regulates *BmCDK20* by competitively binding to miR-3047-z. The upregulation and downregulation of lncRNA41584/miR-3047-z/*BmCDK20* significantly affect the proliferation of BmN cells. The overexpression of lncRNA41584 in BmN cells led to the downregulation of miR-3047-z and the upregulation of *BmCDK20* expression, whereas the opposite occurred upon siRNA knockdown. In the MTT assay, downregulation of lncRNA41584 or *BmCDK20* or upregulation of miR-3047-z led to decreased cell proliferation, whereas upregulation of lncRNA41584 or downregulation of miR-3047-z promoted cell proliferation. Flow cytometry analysis of BmN cells following the knockdown of lncRNA41584 or *BmCDK20* revealed that most cells were arrested in the G1 phase. These results further confirmed the regulatory relationship between lncRNA41584-miR-3047-z-*BmCDK20* and its effects on cell proliferation.

In silkworm experiments, lncRNA41584 continued to regulate miR-3047-z and *BmCDK20*. Upon phenotypic observation, knockdown of lncRNA41584 and overexpression of miR-3047-z resulted in a significant increase in malformed spermatocysts compared with those in the NC control group. Additionally, knockdown of lncRNA41584 led to a decrease in the fertilization rate. Further observation of mating and egg laying behavior in p50 male moths with lncRNA41584 knockdown revealed a significant reduction in fertilization rates and a greater number of unfertilized eggs than those in the control group.

Recent studies have shown that the deletion of factors such as PGC and Dicer1 leads to blocked germ cell development in mice, resulting in a significantly lower number of germ cells than in wild-type or heterozygous mice, ultimately causing infertility [31]. For example, the deletion of Dicer1 leads to multiple cumulative defects during meiosis and postmeiosis, resulting in the loss of functional sperm [32]. Cyclin-dependent kinases (CDKs) play important roles in the cell cycle. *CDK4*, *CDK6*, and *CDK2*, along with their specific cyclin partners, regulate the transition from the G1 phase to the S phase in mammalian cells [33,34]. A lack of *CDK4* leads to defects in cell proliferation and self-sufficiency, whereas *CDK2* is involved in the precise pairing and recombination of sperm cells and telomeres. In various cancer cells, including glioblastomas, liver cancer, ovarian cancer, and colorectal cancer cells, *CDK20* can activate *CDK2* and control the cell cycle progression of cancer cells [21]. In our study, *BmCDK2* was upregulated alongside *BmCDK20* expression, and vice versa. Additionally, the expression of *BmPCNA*, a gene associated with the cell cycle, was consistent with *BmCDK20* expression. These findings suggest that lncRNA41584 regulates testicular cell proliferation through *BmCDK20*.

In conclusion, given the tissue-specific expression of lncRNA41584 in the testes, it likely plays a crucial role in the male sterility traits of the JMS mutant silkworm. lncRNA41584 regulates miR-3047-z and subsequently controls *BmCDK20*, leading to abnormal testicular cell proliferation and ultimately affecting reproduction. This study reveals the links among lncRNAs, cellular processes, and spermatogenesis in silkworms and provides new insights into the potential mechanisms of male sterility in Lepidoptera. It also offers new targets for insect sterility technologies, opening new avenues for pest control.

## 5. Conclusions

lncRNA41584 regulates miR-3047-z and subsequently controls *BmCDK20*, leading to abnormal testicular cell proliferation and ultimately affecting reproduction.

## Figures and Tables

**Figure 1 insects-16-01120-f001:**
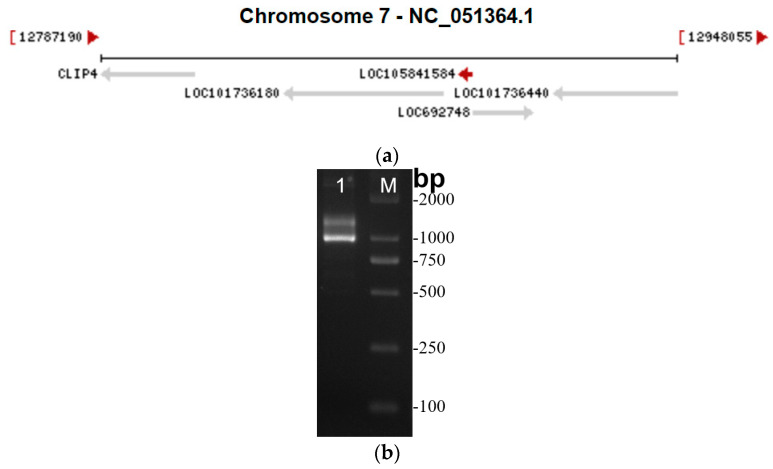
lncRNA41584 is located in the genome at the specific locus of chromosome 7 [25] (**a**); lncRNA41584 full−length amplification PCR product (**b**).

**Figure 2 insects-16-01120-f002:**
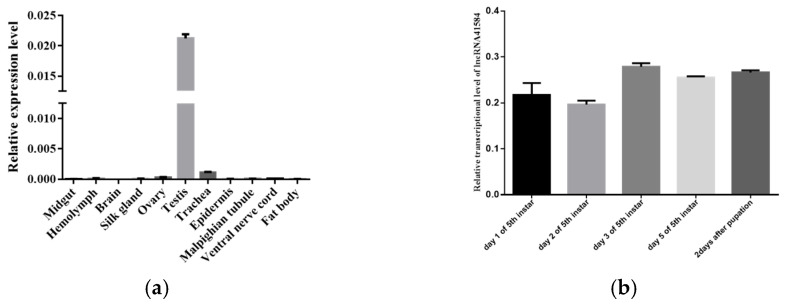
lncRNA41584 transcript levels in 11 tissues of domestic silkworms at day 3 of 5th instar of p50 (**a**); lncRNA41584 transcript levels at different times after 5th instar of p50 (**b**).

**Figure 3 insects-16-01120-f003:**
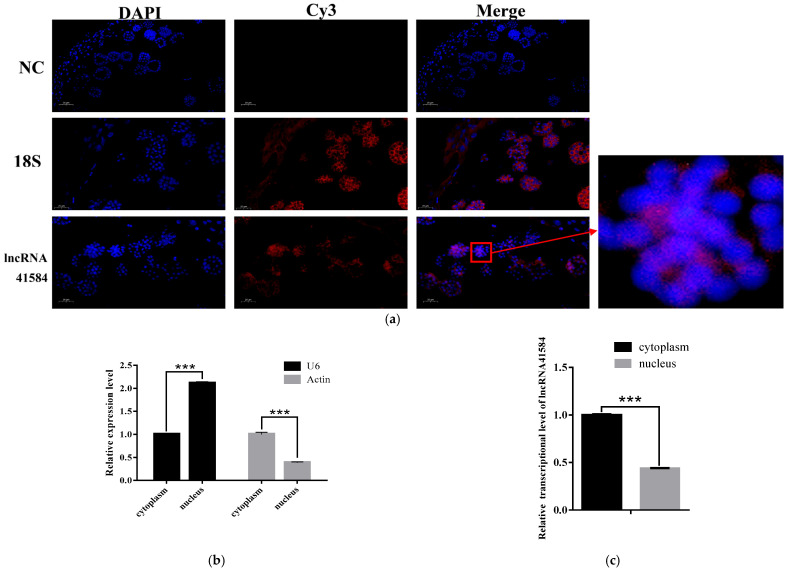
Cellular localization using fluorescence in situ hybridization lncR41584 in nuclei and cytoplasm in sections of day 3 of 5th instar of p50 domestic silkworm spermathecal cells. hybridization probes for lncR41584 and positive control 18S were labeled with Cy3 at the 5′ end, and nuclei were stained with DAPI. Scale bar indicates 20 μm. dAPI, 4,6-diamidino-2-phenylindole, 18S indicates 18S rRNA (**a**); validation after nucleoplasmic separation (**b**); lncRNA41584 transcript levels in nuclei and cytoplasm after nucleoplasmic separation (**c**). (Values are means ± SEM of 3 experiments. *** *p* < 0.001). Significance analysis was conducted with ANOVA and Student’s *t*-test. ANOVA, analysis of variance; NC, normal control; SEM, standard error of the mean.

**Figure 4 insects-16-01120-f004:**
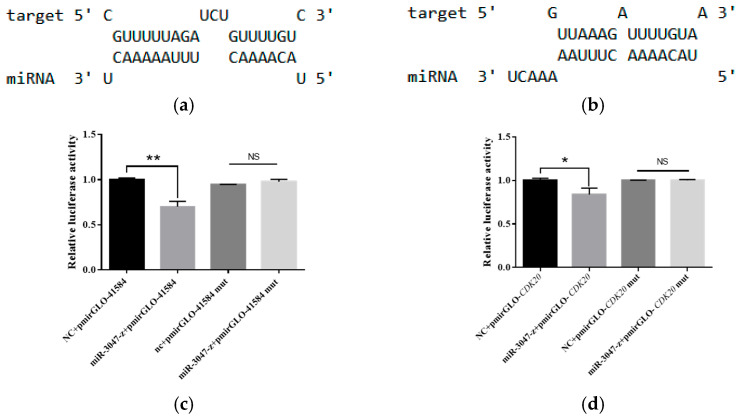
RNAhybrid predicted binding positions of lncRNA41584 to miR-3047-z (**a**); and miR-3047-z to the BmCDK20 3′ UTR (**b**). Targeting of lncRNA41584 to miR-3047-z was verified by luciferase assay (**c**); targeting of miR-3047-z to the BmCDK20-3′UTR fraction was verified by luciferase assay (**d**); NC denotes negative control for miR-3047-z mimics, mut indicates mutant plasmid after point mutation at the binding site. (Values are means ± SEM of 3 experiments. NS *p* > 0.05, * *p* < 0.05, ** *p* < 0.01). Significance analysis was conducted with ANOVA and Student’s *t*-test. ANOVA, analysis of variance; NC, normal control; SEM, standard error of the mean.

**Figure 5 insects-16-01120-f005:**
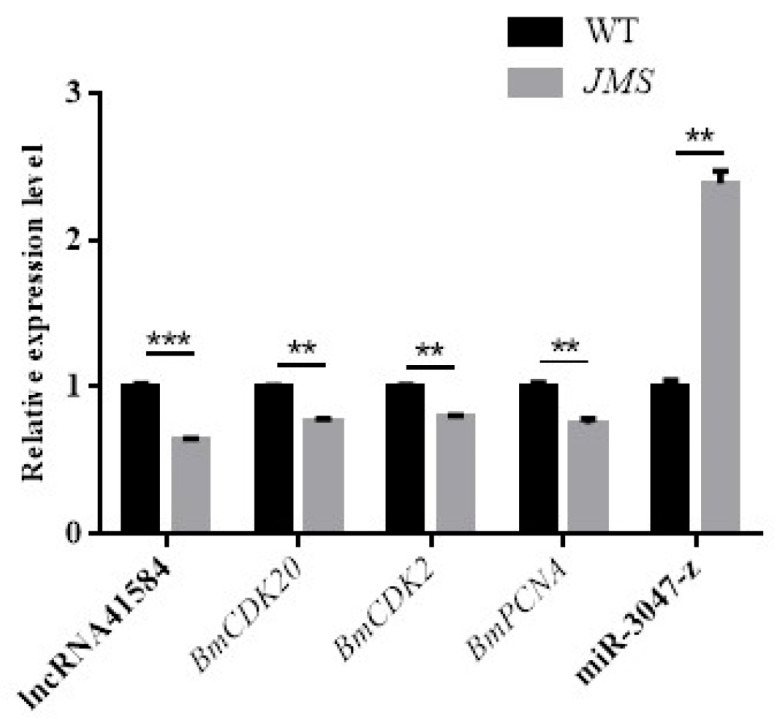
The qPCR analysis of lncR41584/miR-3047-z transcriptional levels and potential regulatory relationship genes in wild-type JN8HUA (WT) and mutant JMS silkworms. (Values are mean ± SEM of 3 experiments. ** *p* < 0.01, *** *p* < 0.001).

**Figure 6 insects-16-01120-f006:**
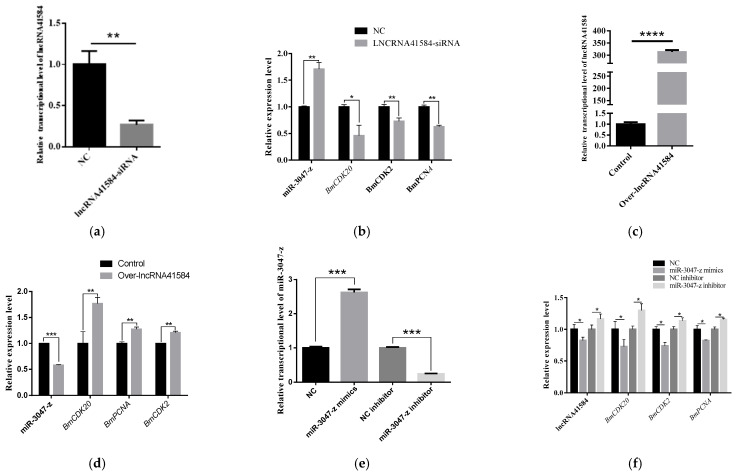
lncRNA41584 regulates cell cycle protein kinase 20 (*BmCDK20*) by binding to miR-3047-z in BmN cell lines. Validation after knockdown of lncRNA41584 (**a**) and miR-3047-z and related mRNA qPCR results after knockdown (**b**); validation after overexpression of lncRNA41584 (**c**), miR-3047-z and related mRNA qPCR results after overexpression (**d**); miR-3047-z knockdown and overexpression validation (**e**); in miR-3047-z 3047-z knockdown and overexpression after lncRNA41584 and associated mRNA qPCR results (**f**). Three independent experiments were performed and each sample was repeated 3 times (values are the mean ± SEM of the 3 experiments. * *p* < 0.05, ** *p* < 0.01, *** *p* < 0.001, **** *p* < 0.0001).

**Figure 7 insects-16-01120-f007:**
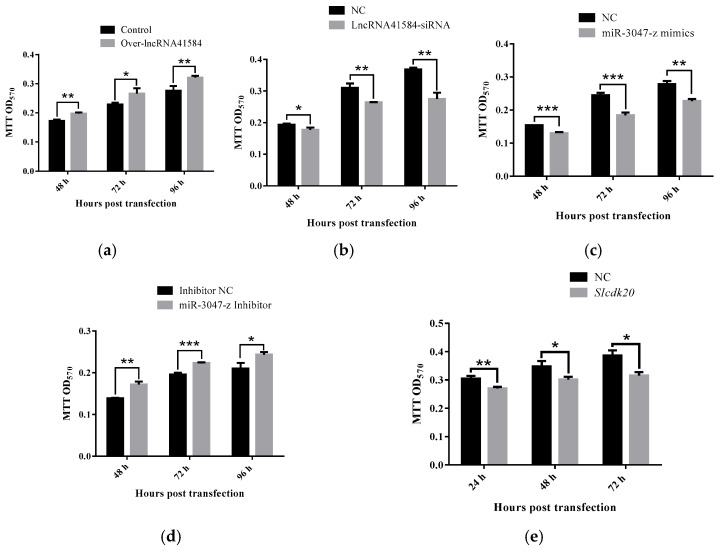
MTT cell proliferation assay after overexpression/knockdown of lncRNA41584/miR-3047-z and knockdown of *BmCDK20* in BmN cells. Detect the absorbance values of MTT reagent added at 48, 72 and 96 h after overexpression of lncRNA41584 (**a**); Knockdown of lncRNA41584 (**b**); Overexpression of miR-3047-z (**c**); Knockdown of miR-3047-z (**d**); Knockdown of *BmCDK20* (**e**). NC indicates the corresponding negative control. Each sample was repeated 3 times (values are the mean ± SEM of the 3 experiments. * *p* < 0.05, ** *p* < 0.01, *** *p* < 0.001).

**Figure 8 insects-16-01120-f008:**
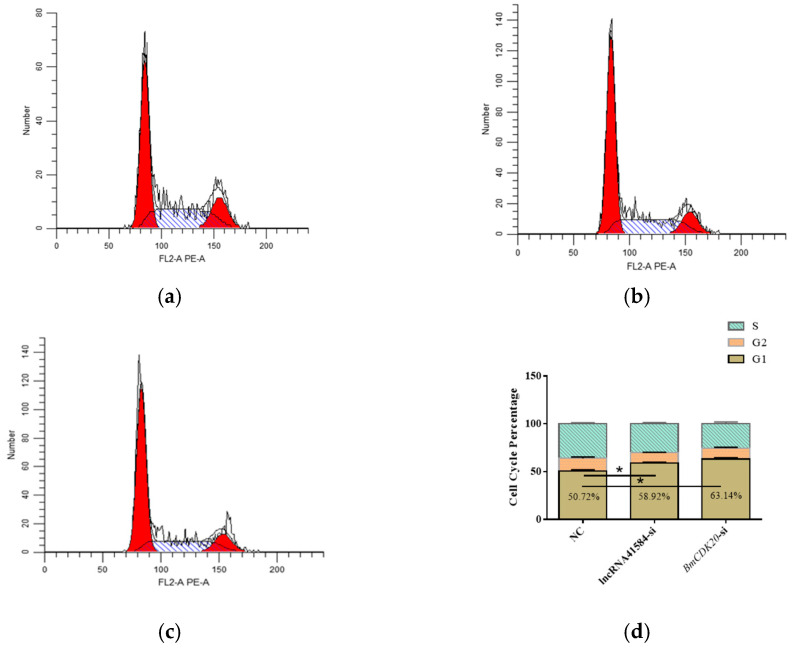
Cell cycle detection by flow cytometry after lncRNA41584 and *BmCDK20* knockdown in BmN cell lines. Negative control (**a**); after lncRNA41584 knockdown (**b**); *BmCDK20* knockdown (**c**); statistics of each cell cycle for three sets of biological replicates (**d**). (values are the mean ± SEM of the 3 experiments. * *p* < 0.05). In Figure (**a**–**c**), the vertical axis represents the relative cell number, and the horizontal axis indicates the fluorescence intensity of PE fluorescent dye. G1 denotes the presynthetic phase of mitosis, S corresponds to the DNA synthesis phase, and G2 signifies the postsynthetic phase.

**Figure 9 insects-16-01120-f009:**
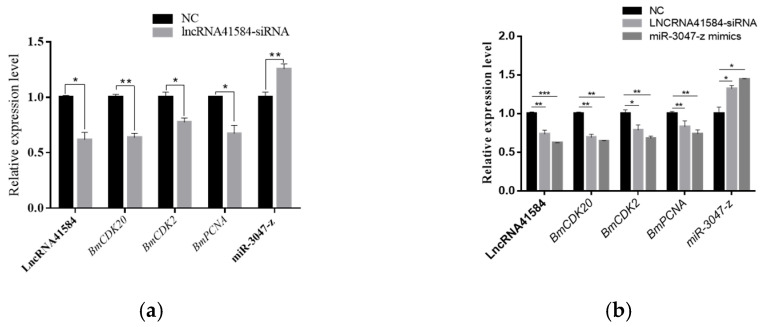
RT-qPCR assay after 48 h of knockdown of lncRNA41584 at third day of fifth instar in p50 (**a**); RT-qPCR assay after 48 h of knockdown of lncRNA41584 and overexpression of miR-3047-z at seventh day of fifth instar in p50 (**b**). Each sample was repeated 3 times (values are the mean ± SEM of the 3 experiments. * *p* < 0.05, ** *p* < 0.01, *** *p* < 0.001).

**Figure 10 insects-16-01120-f010:**
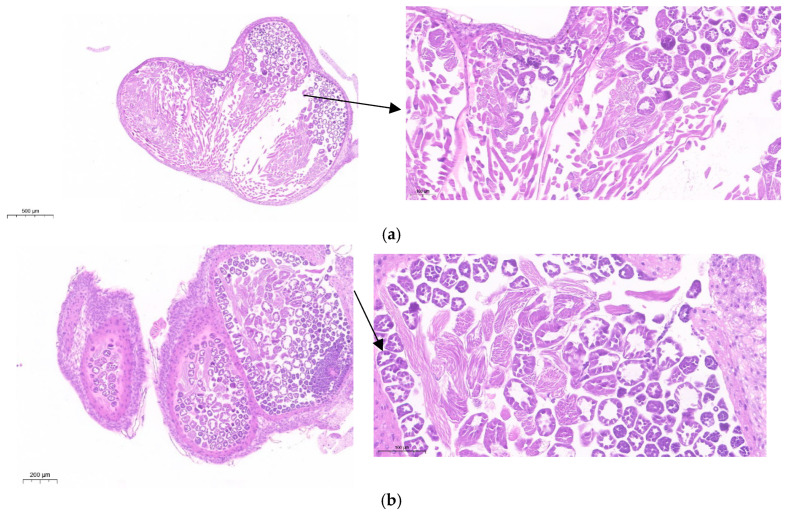
Testicular tissue sections of p50 third day of fifth instar larvae 48 h post siRNA-mediated knockdown of lncRNA41584. Negative control (**a**); knockdown of lncRNA41584 (**b**).

**Figure 11 insects-16-01120-f011:**
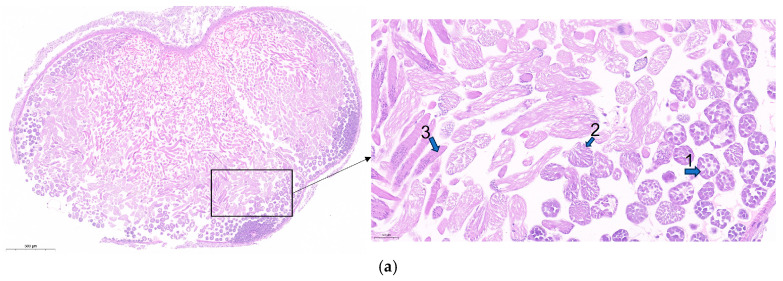
Testicular tissue sections of p50 seventh day of fifth instar larvae 48 h post knockdown of lncRNA41584 or overexpression of miR-3047-z. Negative control (**a**); knockdown of lncRNA41584 (**b**); overexpression of miR-3047-z (**c**). In this Figure, 1 represents primary spermatocytes, 2 shows spermatocysts during spermatogenesis, 3 indicates spermatogonia, and 2′ depicts abnormal spermatocysts during spermatogenesis.

**Figure 12 insects-16-01120-f012:**
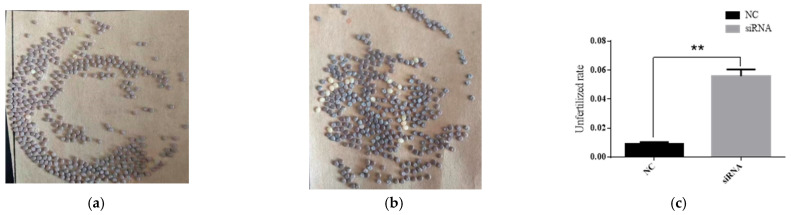
Statistical analysis of fertilization rates in p50 following siRNA-mediated knockdown of lncRNA41584 at seventh day of fifth instar larval stage. Negative control ♂ × p50 wild type (**a**); lncRNA41584-siRNA ♂ × p50 wild type ♀ (**b**); Statistical analysis of unfertilized egg rates following siRNA-mediated knockdown of lncRNA41584 (**c**) (values are the mean ± SEM of the 3 experiments. ** *p* < 0.01).

**Table 1 insects-16-01120-t001:** Main Primers.

Primer	Sequence 5′-3′
BmCDK20-gF	CGTTGGTCGAATCGGTGAAGGA
BmCDK20-gR	GTCAGTTCATGCTCGCATTGGT
lncRNA41584-gF	TTGTGACGGTGGCGTTATG
lncRNA41584-gR	TTCCCGCTTTGTCGCTTGT
Long Primer	CTAATACGACTCACTATAGGGCAAGCAGTGGTATCAACGCAGAGT
Short Primer	CTAATACGACTCACTATAGGGC
lncRNA41584-5race	GACTGAAGCCGTGCCCGTCTATCCAATCTG
lncRNA41584-F	GTAAAGACAATTTGATGTTATCATCTAAAAAAATC
lncRNA41584-R	CGTGGGTTGAATCTGTTTTTAATTATTAC
41584-Cy3	CAAUCUGGUGUAAUGUUAAAU
NC-Cy3	UGCUUUGCACGGUAACGCCUGUUUU
18S-Cy3	CUUCCUUGGAUGUGGUAGCCGUUUC
lncRNA41584-siRNA	CGAUUUCGUUCCGAAAUUATT
miR-3047-z	uACAAAACuuuAAAAACu
BmCDK20-siRNA	GAGAAGAUCCAGCGAGAAATT
BmGAPDH-gF	GTGTCCTCAGACTTCATTGG
BmGAPDH-gR	AATGACTCTGCTGGAATAACC
BmU6-F	CGTATACTAAAATTGGAACGATACAG
BmU6-R	ATTTTGCGTGTCATCCTTGC
BmActin-F	CCGTATGCGAAAGGAAATCA
BmActin-R	TTGGAAGGTAGAGAGGGAGG
41584-GSP-1	GTGTCCCGATGTTTTTGTGACG
41584-GSP-2	TTATATTAGTATTTTATCCGTTTTTAGATC
miR-3047-z RT	GTCGTATCCAGTGCAGGGTCCGAGGTATTCGCACTGGATACGACGTTTTT
miR-3047-z F	GCGCGCGCGACAAAAC
miRNA reverse primer	AGTGCAGGGTCCGAGGTATT
lncRNA41584-Xho1 F	ccgctcgagGTAAAGACAATTTGATG
lncRNA41584-Not1 R	ATTTGCGGCCGCTTTTCGTGGGTTGAATCTG
BmCDK20-xho1 F	CCGCTCGAGCTAAATTCAACTTGCTACAAAAATAATTATG
BmCDK20-Not1 R	ATTTGCGGCCGCACACAAAACTTATAGCCTACTCAG
BmPCNA-gF	TGAATCTAGGCAGCATGTCAA
BmPCNA-gR	TCCTGTGCTTTTATTGTGGCTG

## Data Availability

Data available in a publicly accessible repository. The original data presented in the study are openly available in https://www.ncbi.nlm.nih.gov/ (10 April 2025), GENG ID BankIt2930851.

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
