# Peer review of "The lncRNA41584-miR3047-z-*BmCDK20* ceRNA Regulatory Network Influences Reproductive Development in Male Silkworms (*Bombyx mori*)"

_insects, 2025, doi:10.3390/insects16111120_

Round 1

Reviewer 1 Report

Comments and Suggestions for Authors

LncRNAs are molecules that modulate cellular processes, acting as regulators of gene expression or as templates for miRNA production. This study evaluated the relationship of lncRNA41584 with male sterility in silkworms. Furthermore, a possible pathway of activity for this lncRNA was suggested, given that its expression occurred almost exclusively in the testes and the fertilization rate was lower when lncRNA41584 was silenced.

In addition, I have a few comments for the author's consideration that have been separated by the sections of the manuscript.

TITLE

Lines 1-3: Always provide species authorship according to the rules of the International Code of Zoological Nomenclature (ICZN) for all animal names, both in the Title and in their first citation in the text (please, also check line 47 of the Introduction section).

Line 3: Please, write ‘Bombyx mori’ in italics.

KEYWORDS

Line 44: Please, avoid using the same words found in the Title.

INTRODUCTION

Line 47: Please, add scientific name of the species with respective authorship according to the ICZN rules.

Lines 60-112: The authors conducted a robust survey of lncRNAs and other pathways, such as CDK. I understand that this compilation is not the focus of this article (or perhaps this organization could be included in a supplementary table), or it could be included in a future publishable review.

MATERIAL AND METHODS

Line 129: Do not abbreviate the scientific name at the beginning of a sentence. Please, rewrite the scientific name in full.

Line 138: Please, correct the acronym ‘qRT-PCR’ to ‘RT-qPCR’. Check and standardize throughout the text.

Lines: 156-158: How were the gel extraction and purification, and vector ligation procedures performed? Please, provide the protocol or indicate the kits used.

Line 258: Please, add scientific name of the species with respective authorship according to the ICZN rules.

RESULTS

Lines 291-407: In general, the results are well described and indicated, however, excerpts that point to suggestions or inferences (such as lines 318-320; 342-343) should be removed or moved to the Discussion section.

Lines 3696-399: Excerpt without cohesion with the text. Please, rewrite it.

Line 413: Please, provide the reference of the ladder used.

Lines 443-445: Please better relate the group breakdown to their respective graph and increase the font size of the X and Y axes.

Lines 465-468: Why do the Y-axis scales between figure (a) differ from (b) and (c)?

Lines 485-486: Please, mention in the caption what you want to indicate with the arrows used.

DISCUSSION

Lines 531-553: The summary of results should be in the Conclusion section or at the end of the Discussion.

Lines 571-573: Has lncRNA41584-like been found in another Lepidoptera? Even if not, it would be interesting for the authors to delve deeper and discuss how the results found here can be applied to other studies of the Order, even if they are perspectives.

Reviewer 2 Report

Comments and Suggestions for Authors

Dear authors,

Congratulations on the manuscript entitled The lncRNA41584-miR3047-z-BmCDK20 ceRNA regulatory network influences reproductive development in male silkworms (Bombyx mori). It exhibits great importance not only for the sericulture industry, but also for the biotechnology area. 

Before publication, there are some issues to be addressed:

  1. The details of microinjection should be mentioned, for instance, the final injected volume or if the volume was adjusted according to the larval weight, and so on.
  2. The statement mentioned by the authors also offers new targets for insect sterility technologies, opening new avenues for pest control. or identifies potential new targets for developing insect sterility-based pest control strategies, offering a more environmentally friendly approach to pest management, is not sufficiently supported, as the silkworm, Bombyx mori, is a domesticated species. 
  3. The entire manuscript should be revised for uniform terminology (example:  lncRNA4158, and so on).
  4. The resolution of Figures 3, 6, and 8 should be enhanced.

Best regards

Reviewer 3 Report

Comments and Suggestions for Authors

The study found that the testis-enriched long non-coding RNA lncRNA41584 plays a role in male fertility in silkworms (Bombyx mori). lncRNA41584 acts as a competing endogenous RNA (ceRNA) by sponging miR-3047-z, which in turn upregulates BmCDK20, a cyclin-dependent kinase essential for cell cycle progression. Disruption of this lncRNA–miRNA–mRNA axis led to impaired cell proliferation, G1 phase arrest, malformed spermatocysts, reduced fertilization rates, and increased unfertilized eggs, linking it directly to male sterility. The authors suggested that theses findings could lead to new targets for management strategies of pest insects.

The manuscript was well written, the methods clearly presented, and the results supported the conclusions. I have only a modest number of comments and questions for the authors to consider before publishing:

  1. In this study, the authors used TargetScan to identify miR-3047-z as a potential regulator of both lncRNA41584 and CDK20. Given that microRNAs often have multiple targets, could the authors clarify how many additional mRNAs and lncRNAs were predicted to interact with miR-3047-z? Were the predicted binding affinities for these other targets significantly weaker compared to lncRNA41584 and CDK20? importantly, did the computational analysis reveal any other testis-specific genes among the predicted targets, which might also affect fertility, if perturbed by the lncRNA-miRNA interactions?
  2. Did the miR-3047 bind to only a single site in each of these two targets (the CDK20 mRNA and the lncRNA41584)? Often, miRNAs can have multiple binding sites.
  3. It would be informative to indicate the length of the lncRNA and to explain why two products were generated by RACE. Would the qRT-PCR primers identify both variants?
  4. The images in Figure 11 highlighted evidence of incomplete spermatogenesis, pointing out abnormal development with a few arrows. Is it possible to quantify the extent of normal vs abnormal components? The full cross-section images in the left column are not easily interpreted as the resolution does not permit a viewer to readily identify irregularities.
  5. It is speculated that this study could identify new methods of pest insect control – can this concept be expanded upon? Is this lncRNA, miRNA, and regulated CDK20 axis group of genes conserved in pest lepidopteran insects? In other insect clades. A comment on the likelihood of this regulatory network having similar roles in other insects would be helpful.
